# Human iPSC-Derived Astrocytes: A Powerful Tool to Study Primary Astrocyte Dysfunction in the Pathogenesis of Rare Leukodystrophies

**DOI:** 10.3390/ijms23010274

**Published:** 2021-12-27

**Authors:** Angela Lanciotti, Maria Stefania Brignone, Pompeo Macioce, Sergio Visentin, Elena Ambrosini

**Affiliations:** 1Department of Neuroscience, Istituto Superiore di Sanità, 00169 Rome, Italy; angela.lanciotti@iss.it (A.L.); mariastefania.brignone@iss.it (M.S.B.); pompeo.macioce@iss.it (P.M.); 2National Center for Research and Preclinical and Clinical Evaluation of Drugs, Istituto Superiore di Sanità, 00169 Rome, Italy; sergio.visentin@iss.it

**Keywords:** stem cells, human astrocytes, glial cells, AxD, MLC, VWM, Aicardi–Goutières, 3D models, white matter, myelin

## Abstract

Astrocytes are very versatile cells, endowed with multitasking capacities to ensure brain homeostasis maintenance from brain development to adult life. It has become increasingly evident that astrocytes play a central role in many central nervous system pathologies, not only as regulators of defensive responses against brain insults but also as primary culprits of the disease onset and progression. This is particularly evident in some rare leukodystrophies (LDs) where white matter/myelin deterioration is due to primary astrocyte dysfunctions. Understanding the molecular defects causing these LDs may help clarify astrocyte contribution to myelin formation/maintenance and favor the identification of possible therapeutic targets for LDs and other CNS demyelinating diseases. To date, the pathogenic mechanisms of these LDs are poorly known due to the rarity of the pathological tissue and the failure of the animal models to fully recapitulate the human diseases. Thus, the development of human induced pluripotent stem cells (hiPSC) from patient fibroblasts and their differentiation into astrocytes is a promising approach to overcome these issues. In this review, we discuss the primary role of astrocytes in LD pathogenesis, the experimental models currently available and the advantages, future evolutions, perspectives, and limitations of hiPSC to study pathologies implying astrocyte dysfunctions.

## 1. Introduction

### 1.1. Astrocytes as Playmakers of the Central Nervous System (CNS) Physiopathology

Astrocytes represent the most numerous population of glial cells of the CNS. Due to their multitasking capacities and by establishing close contacts with all the other cellular elements of the CNS, these star shaped cells are the main effectors of brain tissue homeostasis, allowing the maintenance of appropriate environmental conditions required for CNS proper functionality from the first developmental stages to the adult life [1,2]. On the basis of their morphology, astrocytes were originally classified in two different populations: (i) protoplasmic astrocytes, highly branched astrocytes of the grey matter (GM), closely interacting with neuronal synapses and forming the “tripartite synapses”; and (ii) the fibrous astrocytes, a less branched cellular type, mostly present in the white matter (WM) where they contact oligodendrocytes and axons [3,4]. Evidence accumulated in the last 20 years have expanded this initial classification, revealing that astrocytes are a more heterogeneous group of cells showing regional specificities manifested by morphological, molecular and functional differences and whose knowledge is still to be completed [5,6]. By extending their processes and end-feet along blood vessels, perivascular astrocytes also take part to the formation of the blood-brain-barrier (BBB), the anatomic and biochemical barrier facilitating the entry of nutrients and excluding harmful substances into the brain [7,8,9]. Perivascular astrocytes not only participate in BBB formation and maintenance but they also coordinate blood flow with neural activity and brain metabolic needs, forming the so-called neurovascular unit (NVU) [10,11]. In addition, within brain parenchyma, astrocytes share their cytoplasm through gap junction coupling creating a functional reticular system called “astrocyte syncytium”. The syncytium permits astrocytes to exchange signaling molecules, mainly calcium ions, in response to different physiological or pathological stimuli. This intercellular calcium passage generates a dynamic flux of calcium waves that favors cellular communication at long distance, providing astrocytes with a specific form of excitability, even if they are not considered proper excitable cells [12,13]. Due to the intimate relationships between astrocytes and the other cellular elements of the CNS and their capacity to sense and share signals among themselves, astrocytes control essential brain physiological processes. Indeed, astroglia contributes to neuron/synapsis development and function, oligodendrocyte/myelin formation and maintenance, neurotransmitter/ionic homeostasis control, blood–brain barrier formation and biochemical properties and immune response regulation [14,15,16,17]. To exert these complex tasks and in response to physiological signals, astrocytes can undergo reversible variations in morphology and gene expression profile according to CNS specific functional, temporal, and regional needs, a process called “astrocyte activation” [18]. In pathological situations, astrocytes can switch to a reactive state known as “astrogliosis” (or astrocytosis or reactive gliosis), a more complex status characterized by a finely tuned spectrum of morphological and functional changes that include gene expression alterations, release of soluble factors (such as cytokines, chemokines, growth factors, ATP, and other inflammatory mediators), cellular hypertrophy, proliferation, scar formation, and tissue rearrangement [18,19,20]. Although astrocytosis aims at tissue protection, it can reveal harmful if not appropriately controlled [20,21]. Thus, alterations of the strictly regulated tissue homeostasis due to astrocyte dysfunctions can cause widespread detrimental effects on CNS functionality. Indeed, astrocyte defects have been found to contribute to the onset, progression, and resolution of numerous diseases, including trauma, infections, neurodegenerative, neuroinflammatory, neurodevelopmental and neuropsychiatric diseases [22,23,24,25,26]. 

### 1.2. Application of iPSC Technology to Study Rare Neurological Diseases

Clarifying the role of astrocyte in specific CNS diseases opens the possibility to manipulate specific astroglial populations for therapeutic purposes [27]. To this aim, innovative tools for molecular, genetic, morphological, and physiological assessment of astrocyte biology in vivo have been recently developed [28]. Despite these advancements, there are many neurological conditions, particularly among rare diseases (RDs), in which astrocyte involvement is not fully known [29]. According to EU criteria, RDs affect less than 1 out of 2000 individuals. Most RDs are chronic degenerative with neurological manifestations and are life threatening [30]. One of the main bottlenecks hampering RD research progresses is the difficulty to generate relevant experimental models where studying disease molecular pathogenesis. Since pathological brain tissue is not easily accessible, most of these researches have been carried out on animal models, mainly engineered rodents (knock-out or knock-in and cells derived from them), and sometimes zebrafish, to recapitulate the human pathologies [31]. Although these animal models have been essential for the elucidation of some aspects of astrocyte physiopathology, they showed some limits. Compensatory mechanisms, lower genetic variability of the inbred animals, age of disease assessment and environmental conditions may be all involved in the lack of reproducibility of the human disease phenotype [32,33]. Noteworthy, it has become increasingly evident that human astrocytes differ from the rodent counterparts for the expression of a unique set of genes [34], morphological complexity [35], and inflammatory response [36]. Astrocyte functional studies on human in vitro models have been performed on primary fetal or adult astrocytes or, alternatively, on immortalized astrocytoma cell lines, but both of them are not suitable for pathological modeling, due to low availability and the tumoral origin, respectively. In this scenario, hiPSC-development from somatic cell reprogramming (and all the progresses of these techniques, see below) represented one of the most exciting technological breakthroughs made in the last years. The hiPSC are stem cells originally obtained by reprogramming differentiated adult somatic cells (mainly skin fibroblasts), through the introduction of the four Yamanaka transcription factors (OCT4, SOX2, KLF4, and c-MYC) to allow cells to retrieve a pluripotent state [37,38]. Expression of these exogenous factors produces a gradual silencing of the differentiating phenotype markers while it triggers the expression of markers of the pluripotent state. As pluripotent cells, hiPSC have, theoretically, the ability to generate all the cellular types of the body, through the use of specifically optimized methodologies for their culture. From human hiPSC, several cellular types have been induced (i.e., adipocytes, cardiomyocytes, hematopoietic cells, and pancreatic beta cells) including CNS-resident cells [38]. This made it possible to generate brain-derived cells from patients affected by neurological diseases where studying the molecular/cellular defects in a patient-based pathological context [39,40,41]. For these reasons, this technology is particularly attractive to study leukodystrophies (LDs), a group of rare congenital neurologic diseases characterized by abnormal development or destruction of the myelin sheath of the CNS WM [42]. 

## 2. Leukodystrophies: Rare Disease of the CNS Affecting Myelin Structure and Functionality

LDs are heritable, highly disabling disorders of the WM causing cognitive/motor dysfunctions and involving glial cells in the neuropathological process [43,44]. Currently, LDs remain incurable diseases with poor life expectancy and whose treatment is only symptomatic. LDs have relatively high heterogeneity in the molecular mechanisms and cellular elements involved, which make the comprehension of their pathogenesis very challenging [44,45,46]. The majority of LDs have an infancy onset with progressive degeneration of the intellectual and motor capacities and are often fatal. Very few of them can improve over time [44]. To date, about 30 diseases have been classified as LDs [43] but in the last decade the number of LDs linked to specific gene mutations has consistently increased thanks to the improvements of the DNA sequencing technologies and the definition of effective imaging criteria for their diagnosis [42,43,47]. Currently, LD diagnosis follows a consequential step process encompassing a first supposed diagnosis, based on clinical and imaging evidence (by magnetic resonance imaging (MRI)), then a genetic test for confirmation [42]. LD-linked genetic mutations can affect the function of genes expressed ubiquitously or, more specifically, by a single population of glial cells, including astrocytes, oligodendrocytes, and microglial cells leading to structural and functional alterations of myelin of the WM. In the CNS, the myelin sheath is formed by the enwrapping of the oligodendrocyte plasma membrane around the axons, with a single oligodendrocyte forming the myelin sheath of several axons. Myelin forms a protective coating around axons providing high resistance, low capacitance electrical insulation. Its interruption at nodes of Ranvier facilitates the rapid saltatory impulse propagation. Moreover, myelinating oligodendrocytes also support axonal structural integrity and control neuronal energy and metabolism in strict cooperation with astrocytes [48,49]. Alterations of the myelin structure or its biochemical composition, glial cells dysfunctions, or tissue homeostasis imbalance can strongly affect axonal conduction and the functionality of the neural networks. Due to their heterogeneity, LDs were subjected to different methods of classification, that were originally based on the typology of the myelin defects observed [44,45]. Recently, thanks to the advancements in the comprehension of LDs molecular and cellular defects, a new, more comprehensive method for their classification has been proposed based on the glial cell populations mainly involved in the pathological process. Following this rationale, LDs were classified in oligodendrocytopathies (or primary myelin disorders), astrocytopathies, microgliopathies, leuko-axonopathies, and leuko-vasculopathies [44,46]. 

In the present review we will address the clinical/neuropathological features of each single LD caused mainly (or exclusively) by astrocyte dysfunctions (astrocytopathies).

## 3. Astrocytopathies: When Myelin Defects Are Caused by Astrocyte Disfunctions

For a long time, LDs were thought to be caused by mutations in oligodendrocyte specific genes, but the identification of some LDs caused by mutations in astrocyte unique genes, such as the glial fibrillary acidic (*GFAP*) gene in Alexander disease (AxD) and *MLC1* in Megalencephalic leukoencephalopathy with subcortical cyst disease (MLC), completely changed this view. Moreover, in Vanishing White Matter (VWM), where mutations localize in the ubiquitously expressed translation initiation factor *eIF2b* gene, astrocyte dysfunction was found central in disease pathogenesis, leading to classify these three LDs as “astrocytopathies” [50,51,52]. More recently, a predominant role of astrocytes was proposed also in Aicardi–Goutières syndrome, ClC-2-related disease, Oculodentodigital dysplasia (ODDD) and Giant axonal neuropathy (GAN) [52]. A primary/predominant role for astrocytes in LD pathogenesis should not surprise. Indeed, astrocytes participate in myelin development and maintenance by directly regulating oligodendrocyte survival, differentiation, and maturation from oligodendrocyte progenitor cells (OPC). They also control the extracellular ion homeostasis (particularly of K^+^/Na^+^/Ca^++^ ions) in the extracellular milieu during myelin formation [53,54] by syncytium-mediated ion buffering and BBB exchange. Although all these processes were found dysregulated in the above astrocytopathies (Figure 1), the knowledge of the astrocyte-dependent molecular mechanisms leading to myelin deterioration is still elusive. The reprogramming of hiPSC offers the possibility to generate human astrocytes (and other CNS-resident cells) directly from patients and use them as experimental models to clarify LD pathogenesis. These results will foreseeably provide new insights for the identification of possible molecular targets for therapeutic approaches. Some research groups have already generated human pathological astrocytes from LD patients (Table 1) by the hiPSC technique, providing new and unexpected knowledge on LD pathomechanisms. These advancements will be discussed in the next sections, individually for each astrocytopathy. Astrocyte-specific dysfunctions and the experimental models currently available to study each astrocytopathy are summarized in Figure 2, Figure 3, Figure 4, Figure 5, Figure 6, Figure 7 and Figure 8.

### 3.1. Alexander Disease

Alexander disease (AxD, Figure 2) represents the first example of an LD caused by a primary astrocyte dysfunction [68,69] since it is caused by sporadic dominant mutations in the GFAP gene encoding the GFAP protein [68], the astrocyte-specific marker. The disease can manifest in two forms characterized by a different age of onset and severity [70]. An MRI analysis usually shows cerebral WM changes and swelling with extensive demyelination affecting primarily the frontal lobe, and then extending posteriorly. Patients also develop seizures, psychomotor developmental retardation, macrocephaly. Many mutations in the *GFAP* gene have been described spanning the whole protein sequence [69]. GFAP is an intermediate filament protein regulating the morphology and motility of astrocytes, as well as their interaction with oligodendrocytes, and whose expression increases during reactive astrogliosis and CNS injury. *GFAP* gene variants found in AxD patients act as gain of function mutations that abolish protein dimerization leading to the formation of insoluble aggregates, known as Rosenthal fibers (RFs), accumulating into astrocyte enlarged cell bodies [69,71,72]. Accumulation of RFs within the astrocyte cytoplasm causes cellular dysfunction with devastating effects on the CNS, [73]. Many transgenic mice have been engineered carrying GFAP point mutations to mimic the disease-associated variants in humans [69] (Figure 2). Collectively, AxD animal model and the derived cellular models indicated that GFAP protein mutations and RF accumulation lead to proteasome dysfunction, ER stress [72,74], and oxidative stress [75], all contributing to astrocyte activation manifesting mainly with the release and activation of inflammatory stress mediators (MAPK, P38, c-JUN, CD44, Vimentin, GFAP, ALDH111, chemokines CXCL1, CXCL10, CCL2, hyaluronan) [69,72,76]. In addition, decreased levels of gene transcripts encoding proteins involved in ion/water exchanges and neurotransmitter metabolism such as aquaporin-4 (AQP4), connexins (Cx43, Cx30), glutamate transporter-1 (GLT-1), and glutamine synthetase (GS) were observed [77,78]. Although these transgenic mice exhibited the AxD hallmarks (GFAP aggregation and formation of RF), they did not exhibit myelination defects characterizing human patients [76]. A more recent in vivo model of AxD developed in rats showed some degrees of myelin and axonal degeneration, thus emerging as a suitable model for pathogenic studies and possible drug screening [69]. In addition, a mouse model carrying a heterozygous R236H GFAP point mutation as well as a transgene with a *GFAP* promoter to overexpress human GFAP, showed a decrease in the expression of UDP-galactose-ceramide galactosyltransferase (Ugt8), a regulator of myelin membrane synthesis. Accordingly, these animals also expressed lower levels of the basilar myelin protein constituents (Cnp, Mbp, Mog, Mobp, Mag, and Plp1) [79] suggesting this model as a useful tool to study AxD myelination defects. Newly generated zebrafish models revealed instead a useful system to study the early stages of disease [80,81], but, overall, these animal models (Figure 2) did not succeed in clarifying the relationship between diseased astrocytes and oligodendrocyte/myelin defects. To investigate further this issue, several laboratories generated astrocytes and oligodendrocyte progenitor cells (OPCs) from AxD patient-derived hiPSC (Table 1). Using astrocytes differentiated from hiPSC carrying the R88C or R416W pathological variants, Jones et al. [56] found alterations in astrocytic intracellular vesicle trafficking, calcium dynamics, and ATP release. Defects in these pathways were previously observed in transgenic mice expressing the human R239H variant, though the ATP release was changed in the opposite direction. Further investigations are needed to understand these contradictory results that highlight possible differences in the functional properties of human versus murine astrocytes. By introducing the common R239C variant into human embryonic stem cells (ESCs) and inducing astrocyte differentiation, Canals et al. [61] reported a downregulation of the potassium (K) channel Kir4.1 and of the Na, K-ATPase pump suggesting that K^+^ ion imbalance can have a role in AxD pathogenesis (Table 1). In addition, using patient-derived hiPSC, Li et al. [60] examined astrocyte-oligodendrocyte interactions, finding that AxD mutant astrocytes carrying R279C, R239C, and M73K mutations in the *GFAP* reduced the proliferation of co-cultured OPC derived from control hiPSC, causing a myelination defect in a 3D nanofiber culture system (Table 1). The following transcriptome analysis of the hiPSC-derived cells demonstrated the upregulation of genes involved in immune cell activation, pro-inflammatory cytokine signaling, cell proliferation, and cell adhesion and downregulation of genes involved in synapse control and ion transport in AxD astrocytes. Moreover, the authors provided evidence that the inhibition of chitinase-3-like protein 1 (a secreted protein highly expressed by AxD astrocytes and linked to various neuroinflammatory conditions), through the use of a neutralizing antibody or shRNA knockdown of its receptor on OPC, blocked the inhibitory effect of the AxD astrocyte-derived media on OPC proliferation and myelination. In a more recent paper, using hiPSC-derived astrocytes and brain tissue from AxD patients the authors revealed that a site-specific phosphorylation and caspase cleavage of GFAP can be monitored as a marker of disease severity [82].

All these studies demonstrated the possibility to generate human astrocytes that closely resembled AxD pathological astrocytes at the phenotypic, molecular, and functional levels to further clarify disease pathogenesis and identify possible therapeutic targets and disease prognosticmarkers. Indeed, results obtained revealed for the first time the pathological relationships between AxD astrocytes and oligodendrocytes, demonstrating that GFAP defective astrocytes cause oligodendrocyte maturation and myelination deficits (Figure 1 and Figure 2), thus suggesting a possible way of drug screening for this disease.

### 3.2. Megalencephalic Leukoencephalopathy with Subcortical Cysts

Megalencephalic Leukoencephalopathy with subcortical Cysts (MLC, Figure 3) is a very rare LD characterized by macrocephaly, brain edema and subcortical cysts causing progressive deterioration of motor functions, cognitive decline and seizures; symptoms that often worsen after mild trauma or fever [83]. Histological examination of some autopsy and biopsy-derived tissue samples showed the presence of countless vacuoles localized in the outer layers of the myelin sheaths and in perivascular astrocytes [84,85,86,87]. Reactive astrocytes with swollen end-feet and an accumulation of αβ-crystallin, all markers of cellular stress, have been described [87,88]. In about 80% of patients, the disease is caused by recessive mutations in the *MLC1* gene encoding the membrane protein MLC1 [85]. This protein is almost exclusively expressed by astrocyte end-feet contacting the BBB and blood-cerebrospinal fluid barriers and by the Bergmann glia in the cerebellum [89]. About 20% of patients carry recessive or dominant mutations in a second gene, the *GlialCAM* gene [88], encoding an adhesion such as protein (GlialCAM) that facilitates MLC1 expression at astrocyte plasma membrane [90]. Beyond the classic deteriorating phenotype, caused by recessive mutations in the *MLC1* (MLC1 form of the disease) or in *GlialCAM* (MLC2A form) genes, an MLC remitting disease variant was recently observed in patients with dominant mutations in *GlialCAM* (MLC2B) and in some patients with *MLC1* mutations [91]. These observations suggest that the comprehension of MLC molecular pathogenesis and the identification of possible molecular targets may help finding therapeutic strategies to correct the pathological defect or slow down disease progression. To this aim, several transgenic mouse models of MLC have been generated, mainly *MLC1/GlialCAM* KO [92,93,94,95,96], and double KO for both *MLC1* and *GlialCAM* [97]. Moreover, zebrafish lines KO for *MLC1/GlialCAM* or both have been also developed [97,98] (Figure 3). In all these models, increased brain water content and cerebellar WM vacuolation have been reported, although at an advanced age (about 8 months) compared with the human disease. However, mice do not develop brain cysts and motor/cognitive abnormalities characterizing the human disease [99]. Recently, alterations of the gliovascular compartment have been described in *MLC1* KO mice at early post-natal stages suggesting that BBB/NVU deficits anticipate myelin degeneration but the molecular mechanisms underlining these defects and MLC1 role in these processes are still unknown [100]. Astrocytes derived from transgenic animals and human U251 astrocytoma cell lines engineered to express WT or mutant MLC1 revealed that this protein takes part in a multiprotein complex including several proteins involved in water/ion homeostasis at the BBB compartment (GlialCAM, ClC-2, Na,K-ATPase, Kir4.1, AQP4, TRPV4, LRRC8, V-ATPase, Cx43, and Cav-1), [99,101,102,103,104,105,106,107] and that MLC1/GlialCAM mutations affect astrocytic ion/fluid exchanges and volume rescue following cell swelling induced by hyposmotic stress [92,108], (Figure 1 and Figure 3). It was hypothesized that MLC1 protein mutations affect the capacity of astrocytes to buffer K^+^ ions released during the axonal activity through the astrocyte syncytium and finally to blood circulation (along with water and the counter ion chloride), provoking osmotic disturbances. Epileptic seizures, myelin compaction defects, astrocyte swelling and vacuolation would be the consequences of these K^+^ disturbances [93,109,110]. Since astrocyte swelling also occurs during astrocyte activation (astrogliosis), we also hypothesized that MLC1 mutations may lead to an abnormal/prolonged astrocyte activation and swelling in response to alteration of homeostatic conditions. Consistent with this, we reported the upregulation of endogenous MLC1 in brain inflammatory conditions (AD, CJD, PD, and MS). Interestingly, in different cellular models (human astrocytoma cells and *MLC1* KO primary astrocytes) MLC1-inhibits specific intracellular signaling cascades responsible for astrocyte activation and proliferation following brain inflammation/stress (pEGFR/pERK/pNF-kB/PLCγ/pSTAT3), [107,111,112]. Reactive astrocytes found in MLC brain biopsies and aggravation of patient conditions after fever or mild trauma strongly support this hypothesis. These results also suggest that targeting these molecular pathways with suitable drugs can correct MLC astrocyte defects, as already experimented with an ERK inhibitor in a cellular in vitro model [105]. Evidence that *MLC1* mutations affect astrocyte morphology and motility through actin remodeling, another intracellular process controlled by ERK signaling cascade, has been also provided [113]. Further investigations are thus needed to piece together a more complete outline of MLC1 functions and MLC pathogenesis. To this aim, we recently differentiated human astrocytes from hiPSC lines derived from controls and MLC patients carrying missense and splice site mutations. We used a differentiation protocol to obtain human astrocytes from hiPSC in 4/6 weeks (see Table 1 for details). Preliminary biochemical and proteomic experiments aimed at characterizing the pathological phenotype of these cells confirmed the abnormal activation of EGFR/ERK pathways and revealed new MLC defective processes similar to those described in the other astrocytopathies (AxD and VWM), such as oxidative damage, mitochondrial dysregulation and endoplasmic reticulum stress (manuscript in preparation). These data may open new avenues for the comprehension of astrocyte role in LD and for the development of possible therapeutic strategies for MLC disease.

### 3.3. Vanishing White Matter

Vanishing white matter (VWM, Figure 4), also called Childhood Ataxia with Central Nervous System Hypomyelination (CACH), is one of the most severe and prevalent leukodystrophies [114]. VWM is characterized by chronic progressive neurological deterioration, cerebellar ataxia and optic atrophy, where stresses including febrile infections, acute fright and minor head trauma provoke an acute deterioration of patient conditions [115]. The disease affects individuals from prenatal ages to senescence, but is most prevalent in young children, with clinical severity that is inversely related to the age at onset [116]. MRI shows a bilateral, diffuse, and symmetric involvement of the cerebral WM, with myelin thinning, myelin vacuolation and intramyelin edema, tissue rarefaction and cavitation [117]. Over the clinical course, the cerebral WM typically progresses to cystic degeneration and is eventually replaced by fluid with signal features very similar to those of cerebrospinal fluid [118,119]. Genetically, VWM disease is caused by recessive mutations in any of the five genes encoding subunits (1–5) of the eukaryotic translation initiation factor 2B (*eIF2B*) complex [120,121]. EIF2B is indispensable for the initiation of translation of mRNAs into proteins and it also orchestrates the integrated stress response (ISR), an evolutionarily conserved intracellular signaling network activated in response to cell stress that helps cells to restore their internal balance by reprogramming gene expression [122,123]. From the seminal study by Dietrich et al. [124], performed on post mortem brain tissues, which revealed astrocyte specific impairment in VWM, additional studies carried on patient-derived tissues found that astrocytes were reduced in number while OPC were markedly increased in VWM brain. In addition, VWM astrocytes showed blunt processes and a dysmorphic immature phenotype, since they expressed the astrocyte immature markers GFAPδ isoform, vimentin, αβ-crystallin, and nestin but not astrocyte maturation markers such as GFAPα and S100β. Accumulation of hyaluronan, an extracellular matrix molecule produced by astrocytes and known to inhibit OPC maturation, was also observed in the brain of VWM patients, a condition thought to favor alteration of OPC/oligodendrocyte ratio (for a comprehensive review see Reference [52] and references therein). Another pathological aspect of VWM is the absence of astrogliosis and microglia activation, that it causes an impaired secretion of cytokines supporting OPC differentiation, oligodendrocyte survival, and myelin formation [125]. Although these studies can reflect the pathological features at tissue and cellular level, the scarcity of human pathological material does not allow extensive molecular/biochemical investigations aimed at deciphering the link between eIF2B protein mutations and astrocyte/oligodendrocyte molecular defects. Thus, several VWM mouse models were developed during this time (Figure 4). Mouse knock-in models were generated carrying the point mutations R132H and R191H in *eIF2B5*, the R484W in *eIF2B4*, and the R191H mutation in *eIF2Bε* [126,127,128,129,130] (Figure 4). Recently, a spontaneous point mutation in *eIF2B5* (I98M) was identified in a mouse strain with a small body, abnormal gait, infertility, epileptic seizures, and a shortened lifespan [125]. Different zebrafish models of VWM have been also established and used for proteomic studies [131,132]. All these animal models allowed to recapitulate some key aspects of human disease (increased morbidity, mortality, altered myelination, and impaired motor behavior) and identify the over activation of unfolded protein reaction (UPR), mitochondrial dysfunctions, and glial maturation defects such as molecular defective processes in VWM [115,128,133,134,135]. However, to date, the mouse model phenotype and VWM patient features do not completely overlap, and disease molecular pathogenesis is still elusive. Then, hiPSC models were recently established with the aim of studying VWM in a human-based pathological model and providing a platform for future drug screening (Table 1). Zhou et al. [64], generated VWM hiPSC lines from the fibroblast of two VWM patients carrying compound heterozygous mutations, such as G386V missense mutation and 610–613 deletion in *eIF2B5* and G47E and I346T mutation in *eIF2B3*, respectively. Astrocytes differentiated from neural stem cells (NSCs) expressed the mature astrocytic markers GFAP and S100β and displayed a dysmorphic phenotype manifested as relatively shorter processes with significant increased δ-GFAP and αβ-crystallin. The parallel characterization of NSC, neurons, and oligodendrocytes, differentiated from the same hiPSC showed no alteration in the differentiation capacity, morphology, and apoptosis compared with control cells, further reinforcing the concept that astrocytes play a central role in the pathogenesis of VWM. By inducing hiPSC from fibroblasts of two VWM patients carrying *eIF2B5* mutations (1484A>G and 806G>A gene mutations, respectively) Leferink et al. [63] recently generated human GM and WM astrocyte populations. Importantly, using differentiation media containing either CNTF (WM) or fetal bovine serum (GM), they obtained distinct astrocyte subtypes differing in morphological features and gene expression profiles in vitro (Table 1). This study revealed that patient-derived WM astrocytes, but not GM ones, have intrinsic defects leading to the inhibition of mouse OPC maturation in co-culture system and that WM astrocytes exhibited additional, human specific, phenotypes that were not observed in corresponding cells derived from mouse iPSC. Overall, these findings show for the first time that hiPSC can be used to study intrinsic differences between human astrocyte subtypes, and to identify shared disease mechanisms between species.

### 3.4. Aicardi–Goutières Syndrome

Aicardi–Goutières syndrome (AGS, Figure 5) is a complex inherited autoimmune pediatric disorder with intellectual and physical problems and poor prognosis [136], associated with defects in a subset of genes involved in DNA repair mechanisms, cell cycle progression and regulation [128]. AGS consists so far of seven subtypes, *AGS1* to 7, presenting mutations in the *TREX1, RNASEH2A/B/C, SAMHD1, ADAR1*, and *IFIH1* genes, respectively [136,137,138,139]. Most of the mutations are autosomal recessive, leading to protein loss of function, but dominant inheritance is described in patients carrying gain of function mutations in the *IFIH1* gene [140]. Mutations affect RNA/DNA metabolism, triggering an autoimmune response with an increase in cerebral IFN-α production whose inappropriate stimulation can lead to a severe inflammatory disease. Onset of AGS can occur as early as birth and is characterized by encephalopathy, progressively resulting in a loss of motor and cognitive skills, spasticity, dystonia, and acquired microcephaly. Moreover, necrotic skin lesions (chilblains), and hepatosplenomegaly are typical non-neurological signs [138]. Neuroimaging shows diffuse intracranial calcifications, cortical atrophy, and deep WM hypodensities, while typical neuropathological findings include myelin lack, vascular changes, infarctions, and calcifications [141]. Pathological signs are associated with increased type I interferon (IFN-α) activity and lymphocytosis in the cerebrospinal fluid and serum and activation of IFN-related downstream signaling [142]. Astrocytes are the major source of IFN-α in CNS; thus, they were recently proposed as the main players in AGS pathogenesis [44], though their contribution to the myelin/oligodendrocyte damage is still not clarified [52,143]. Current hypotheses on astrocyte-driven AGS myelin deficits are that mutated, suffering astrocytes are no longer able to support survival/proliferation and myelination capacity of oligodendrocytes and that mutated, activated astrocytes negatively influence OPC maturation due to the detrimental release of IFN-α and inflammatory cytokines [139]. To study AGS pathogenesis, many cellular and animal models were developed (Figure 5), including several null mice for the genes involved in AGS (KO), mice expressing the human mutations (KIN) (*TREX1* KO and KIN, *RNASEH2A/B/C* KO, *RNASEH2A/B* KIN, *SAMHD1* KO, *ADAR1* KO, *IFIH1* KIN) and some zebrafish models (*SAMDH1* and *ADAR1* KO). However, many of these models showed embryonic lethality, while the other did not have an overt brain phenotype ([33] and references therein) [143]. Indeed, lab mice live in relatively sterile environments and lack the initial viral stimulus that may trigger the type I interferon response, condition that can explain the absence of neuroinflammation in these animals [33]. Understanding the contribution of astrocytes to oligodendrocyte dysfunction through hiPSC technology would help reveal mechanisms underlying the disease and suggest possible targets of astrocyte specific therapy in AGS. To this aim, Thomas et al. [67] used hiPSC to dissect the relative contribution of neurons and glia to AGS (Table 1). They generated a *TREX1*-deficient cell line by inducing pluripotency in fibroblasts derived from an AGS patient homozygous for the stereotypical V201D mutation in *TREX1* and differentiated them in neural cells, astrocytes and neurons, and investigated the phenotype of cerebral organoids with cortical fate. In this human AGS model, they observed a neurotoxic response to the abundant extrachromosomal nucleic acids and found that *TREX1*-deficient astrocytes further contributed to neurotoxicity through increased type I IFN secretion. Since in this model reverse transcriptase inhibitors rescued the neurotoxicity, the authors suggest anti-retrovirals as a potentially promising therapy for AGS and related disorders. In order to create an innovative in vitro model to investigate mechanisms of drugs potentially effective in AGS, Genova et al. [144] set up an in vitro assay to study the cytotoxicity effects of a panel of immunomodulators, including medications used to treat AGS on hiPSC derived from fibroblasts of three pediatric patients. By employing hiPSC technology for modeling AGS1, AGS2, and AGS7, they established an innovative in vitro model of AGS, useful to investigate mechanisms of drugs potentially effective in these pathologies (Table 1). Recently, Ferraro et al. [145] generated and characterized three isogenic hiPSC clones carrying a compound heterozygous mutation in TREX1 from fibroblasts of a 5 years of age male affected by AGS1 (Table 1). The same research team also generated and characterized three isogenic hiPSC clones derived from fibroblasts of a 10 years of age female affected by AGS2 due to a homozygous mutation in *RNaseH2B* [146]. Three isogenic hiPSC clones from a *IFIH1* mutated 14 years of age male and proposed as an advanced in vitro model for AGS7 were also developed [147]. Three induced pluripotent stem cell lines from a patient with a deletion of coding exons 14 and 15 of the *SAMHD1* gene were described by Fuchs et al. [148]. All these cell lines (Table 1) will constitute a significant tool to investigate the role of specific mutations in AGS pathogenesis.

### 3.5. Chloride Ion Channel 2 (CLC-2) Related Leukoencephalopathy with Intramyelinic Oedema

ClC-2-related disease (Figure 6) is similar to MLC in terms of pathophysiology, although it is caused by recessive mutations in the *CLCN2* gene that encodes for the ClC-2 channel protein [149]. Patients carrying mutations in ClC-2 manifest a neurological phenotype with variable clinical features including cerebellar ataxia, spasticity, visual field defects, cognitive defects, and headaches. An MRI showed possible myelin vacuolation that was confined to specific WM structures in adult patients, and more diffuse in pediatric patients that do not experience epilepsy [149]. ClC-2 is a widely expressed member of the CLC gene family of voltage-gated chloride (Cl-) channels and transporters [150]; it is expressed at the plasma membrane and it is activated upon membrane hyperpolarization, showing a slow inward rectifying current. Although it is activated also by hypotonic-induced cell swelling, it seems not playing a role in cell volume regulation [151]. Similar to MLC1, ClC-2 chloride channels are mainly located in the astrocytic end-feet enwrapping blood vessels. It is thought that MLC1, GlialCAM, and ClC-2 form ternary complexes in glial membranes, and loss of either GlialCAM or MLC1 changed localization and reduced ClC-2 expression in glia [92,152], suggesting a loss of ClC-2 function as a common factor in MLC, but whether the loss of ClC-2 “opening” by mutated GlialCAM contributes to the pathology remains elusive. To explain the degenerative phenotypes, it was originally proposed that ClC-2 regulates the extracellular ion homeostasis in the clefts between cells [152,153]. To elucidate the physiological role of ClC-2 several ClC-2 KO mice were generated, with ClC-2 constitutive or inducible deletion [154]. These mice display male infertility and blindness and develop LDs with myelin vacuolation. By using these animal models, it was recently found that LDs were fully developed only when ClC-2 was disrupted in both astrocytes and oligodendrocytes and that ClC-2 is crucial for extracellular ion homeostasis in various tissues [155]. The role of ClC-2 has not been investigated as extensively as that of other anion channels. To date, there is only one study taking advantage of hiPSC technology, in which they analyzed the effect of Lubiprostone (LBP), a novel chloride channel opener of ClC-2 and CFTR (cystic fibrosis transmembrane conductance regulator) on mouse-hiPSCs-derived cardiomyocytes [156].

### 3.6. Oculodentodigital Dysplasia

Oculodentodigital dysplasia (ODDD, Figure 7) is a rare, autosomal dominant disorder caused by mutations in the gap junction protein alpha-1 (*GJA1*) gene encoding the gap-junction hemichannel connexin 43 (Cx43). Since Cx43 is abundantly expressed in astrocytes [157], ODDD is regarded as an astrocytopathy [44,158]. This rare disorder affects several organs and tissues causing mostly skeletal anomalies such as syndactyly, broad long bones and particularly, craniofacial abnormalities, with typical dental and ocular anomalies. Neurologic manifestations occur in approximately 30% of patients, and leukodystrophy and MRI anomalies of GM structures was observed in individual cases [159,160]. Due to the astrocytic trophic support, which needs functional gap junction connections, an altered Cx43 expression on astrocytes might influence astroglial crosstalk at several levels, for example by causing unbalanced ionic concentrations [161], accumulation of toxic substances and changes in the immune response [162]. Other tissues expressing Cx43 appear to be unaffected, thanks to the co-expression of other connexin family members. More than 80 different mutations in the *GJA1* gene are associated with ODDD, affecting different domains of the Cx43 molecule and resulting in a variety of neurological symptoms [163]. To date, some mouse models of ODDD have been developed that carry mutations in Cx43, protein but the neurological phenotype was not been characterized [164]. Efforts to understand ODDD pathogenesis and the overall role of Cx43 during development require human-oriented models and tools for studying the underlying connexin-based cellular events. In vitro cellular systems expressing Cx43 pathological mutations in HeLa cells revealed gap junction assembly defects compared with Cx43-WT expressing cells [165]. Esseltine et al. [166] for the first time generated hiPSCs from a patient with ODDD carrying the V216L mutation, reporting that they exhibited reduced Cx43 mRNA and protein amounts, and reduced channel function. Moreover, osteogenic and chondrogenic differentiation of ODDD hiPSC were altered. This study highlights the role of Cx43 during osteoblast and chondrocyte differentiation, founding a potential mechanism to explain how the alteration of cell lineages involved in bone and cartilage development are established by ODDD-associated Cx43 mutations. Although, to date, the consequences of these mutations on oligodendrocyte biology and myelin formation/maintenance are not known, it is conceivable that Cx43 mutations induces alterations of astroglial syncytium properties and/or of astrocyte/OPC/oligodendrocyte relationships occurring through Cx43/Cx47 interactions [52].

### 3.7. Giant Axonal Neuropathy

Giant axonal neuropathy (GAN, Figure 8) is a progressive neurodegenerative disease caused by recessive mutations in the *GAN* gene encoding the ubiquitously expressed protein gigaxonin [167], whose main function is to facilitate degradation of intermediate filament proteins. Similarly to AxD, GAN is characterized by excessive accumulation of GFAP within astrocytes, in addition to neurofilaments (NF-L) and peripherin (PRPH) in neurons; it is therefore classified as astrocytopathy, not only a leuko-axonopathy [45]. Patients with GAN present in their first decade loss of motor and sensory function and their life expectancy is <30 years; currently, there is no treatment for these patients. Given the inaccessibility of primary human neurons, hiPSC are an appealing alternative to model GAN pathology. Although in the available GAN literature so far studies on hiPSC-derived astrocytes are missing, Johnson-Kerner et al. [168] found that hiPSC-derived spinal motor neurons generated from three GAN patients displayed the NF-L and PRPH aggregates characteristic of the early stages of the disease. They demonstrated that the GAN phenotypes can be rescued by gigaxonin replacement and that GAN hiPSC faithfully recapitulate them, proving to be valuable tools to identify therapeutic targets to treat this disease. The overexpression of gigaxonin in primary rat astrocytes revealed its involvement in GFAP degradation through the proteasome pathway [169] giving a possible explanation of the molecular defective pathways in GAN astrocytes. However, knowledge of the causative relationship between GAN defective astrocytes and motor neuron damage is still lacking, a gap that may be filled by the setting up of a hiPSC based pathological model.
Figure 2Phenotype, defective cellular processes, and available pathological models of Alexander diseases.
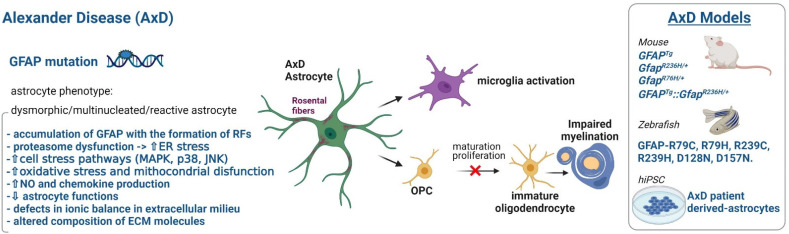

Figure 3Astrocyte phenotype, defective cellular processes, and available pathological models of MLC disease.
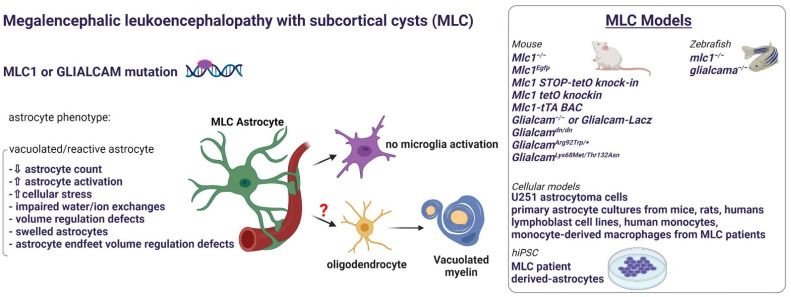

Figure 4Astrocyte phenotype, defective cellular processes and available pathological models of VWM disease.
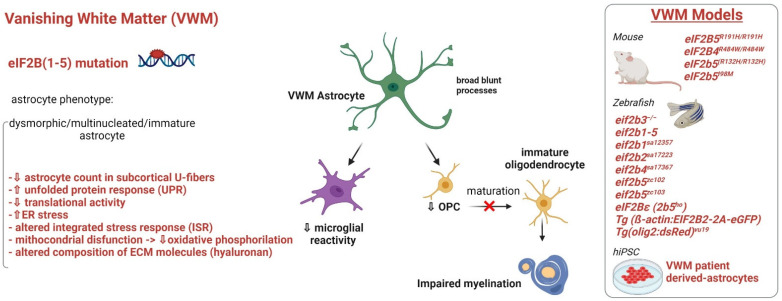

Figure 5Astrocyte phenotype, defective cellular processes and available pathological models of AGS disease.
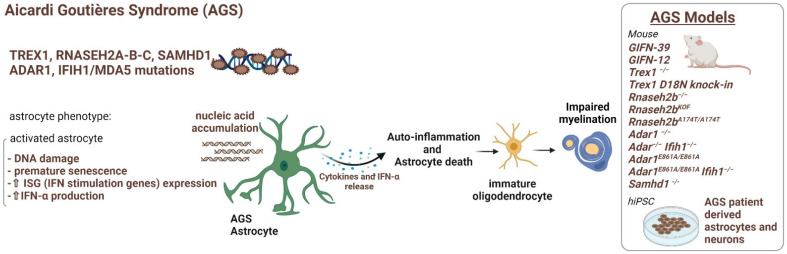

Figure 6Astrocyte phenotype, defective cellular processes and available pathological models of CLC-2 related leukodystrophy.
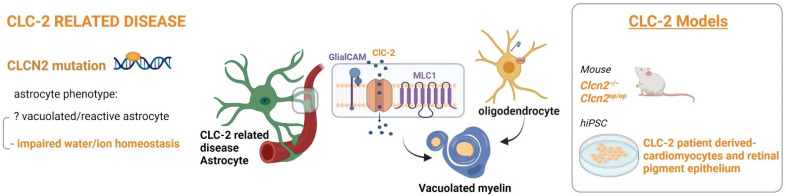

Figure 7Astrocyte phenotype, defective cellular processes and available pathological models of ODDD.
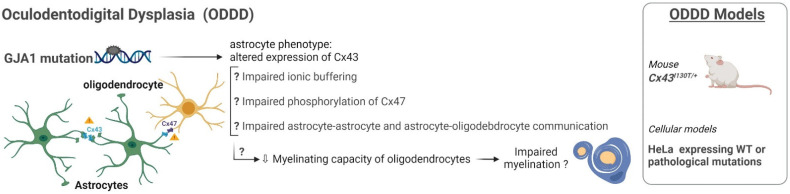

Figure 8Astrocyte phenotype, defective cellular processes and available pathological models of *GAN*.
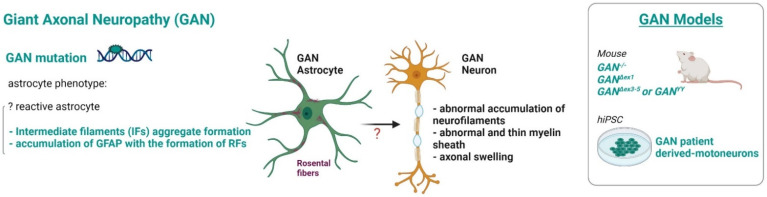


## 4. Advancements of hiPSC-Derived Astrocyte Differentiation Methodologies

In the last decade, several methodologies have been developed to differentiate efficiently astrocytes from human hiPSC. Classically, astrocyte differentiation procedures followed 3 major steps: (i) conversion of undifferentiated hiPSC into neuroepithelial cells forming embryoid bodies (EB) and rosette structures; (ii) induction of neural stem cells/astroglial progenitors (either in adhesion or suspension cultures) with or without morphogens and (iii) terminal differentiation and maturation of human astrocytes. Following this track, Krencick and collaborators [55] developed a chemically defined differentiation system to obtain astroglial progenitors and immature astrocytes from hiPSC in a 6-month period. This methodology was mainly based on the use of FGF2/EGF growth factors and the generation of rosette-forming neuroepithelial cells induced into astroglial progenitors with the administration of morphogens (retinoic acid/sonic hedgehog). Progenitor spheres were then expanded as free-floating clusters or “astrospheres” and finally differentiated into immature astrocytes by dissociating them into single cells that were cultured in the presence of growth factors. To favor astrocyte maturation, in the last differentiation phase, a 6-day stimulation with neural medium supplemented with CNTF was performed. These procedures allowed to obtain large quantities of a uniform population of astrocyte progenitors/immature astrocytes (>90% GFAP+) with the functional hallmarks of primary astrocytes, including the response to glutamate, propagation of calcium waves, promotion of synaptogenesis and participation to BBB formation after transplantation. Following this initial work, the methodologies were optimized with the aim of reducing time and increasing the efficiency of mature astrocytes differentiation, and possibly inducing astrocyte subtypes with regional specification. By using a similar chemical-defined differentiation system, and generating neurosphere as intermediates, other protocols were developed with changes in the type and duration of growth factor supplementation (mainly CNTF, LIF and BMP) in the final differentiation stages. Moreover, other modifications included the use of commercially available medium for NSC stimulation directly from hiPSC, removal or addition of serum in the culture media and use of feeder layers/matrix components for astrocyte induction and growth [63,170,171]. These variations allowed to reduce the differentiation period to 2–3 months, and at the same time, increase the proportion of mature GFAP/S100β-positive astrocytes. More recently, Lundin et al. [172] adapted previously published protocols to obtain, in about 28 days, hiPSC derived astroglia through long-term neuroepithelial-like stem (ltNES) cells, a stable phenotypic and cryo-preservable intermediate model. ltNES can be generated from multiple ESC and hiPSC lines by using well defined culture conditions ensuring high reproducibility across the lines, a very important feature for their direct application in pharmacological high-throughput screenings.

A conceptual different strategy based on transcription-factor programming was lately developed to rapidly and efficiently differentiate functional astrocytes directly from hiPSC. These new strategies foresee to induce in hiPSC the exogenous expression of specific transcription factors, particularly NFIA/B and SOX9, two transcription factors crucial to stimulate gliogenesis during neural development [173]. By using CRISPR/Cas9-mediated inducible expression of NFIA or NFIA plus SOX9 in hPSC (both ES and iPS cells), Li et al. [174] developed a method to efficiently obtain 70%/80% GFAP^+^ astrocytes (also with regional identities) in 4–7 weeks following transcription factors induction. Similarly, Canals and collaborators [61], by means of a lentiviral transfection system, obtained astrocytes from ES and hiPSC after 21 days from NFIA/NFIB and SOX9 activation. In both cases, the differentiated cells displayed typical markers and functional properties of astrocytes, such as glutamate uptake, calcium wave propagation and support to synaptic formation. Recently, functionally mature astrocytes performing glutamate uptake, activating calcium responses and secreting astrocyte-specific cytokines were obtained by using an inducible expression system (SOX9 under the control of the TET-ON promoter) to transient express SOX9 for 6 days in hiPSC-derived neural progenitor cells [175]. In addition, the transient expression of NFIA was found sufficient to trigger the glial competency of hiPSC-derived NSC within 5 days, favoring astrocyte differentiation in the presence of glial-promoting factors. These astrocytes exhibited neuroprotective capacities, activate calcium transients in response to appropriate stimuli and engraft in the adult mouse brain [176]. Overall, these studies demonstrate that SOX9/NFIA expression can generate at high speed functional, region-specific astrocytes on demand.

To overcome the difficulties of cell type heterogeneity during the differentiation process from iPSCs to astrocytes, two different groups genetically modified the iPSC (or ESC) to obtain astrocyte specific reporter cell lines expressing the red/green fluorescent proteins (RFP and eGFP) under the control of the astrocyte-specific GFAP promoter [177,178]. GFAP-driven expression of fluorescent proteins allowed for the enrichment of the differentiated iPSC-derived astrocytes using fluorescent-activated cell sorter (FACS). The homogenous population of purified astrocytes so obtained showed astrocyte marker expression (NF1A, Cx43, GLAST, GS, CD44, and ALDH1L1) [177,178] and functional properties of mature astrocytes, as shown by the ability to respond to neuronal regulation of glutamate transport activity [178].

In parallel to hiPSC-based models, a new approach aimed at building models of the nervous system from human material recently emerged and is gaining growing attention. It relies on the possibility of switching from the phenotype of an already fully differentiated cell to a different (and not necessarily closely related) cell phenotype through a direct conversion procedure based on the use of transcription factors and excluding the undifferentiated state of hiPSC [179]. This process, called “transdifferentiation”, was first applied by Caiazzo and collaborators [180] to generate astrocytes comparable to native brain astrocytes from embryonic and postnatal mouse fibroblasts. The procedure, recently optimized [181], was based on the expression of NFIA, NFIB, and SOX9 transcription factors. Soon after, defined sets of chemical inducers were used to reach the same goal without the use of transcription factors. Since then, several different sets of inducers were tested to improve the standardization and reproducibility of the protocols, with variable success. In brief, the pools of inducers must include epigenetic modulators, such as histone deacetylase or DNA/methyltransferase inhibitors, metabolism modulators, Wnt pathway activators and TGF-β inhibitors. This methodology allowed to obtain astrocytes, oligodendrocytes, and neurons, also including different neurotransmitter-specific subtypes of neurons (glutamatergic, dopaminergic, and gabaergic; reviewed by Masserdotti et al. [182]. One of the most interesting features of the transdifferentiated cells compared with the hiPSC-derived ones, is that the resulting cells maintain the original epigenetic signature acquired through life of the donator subject, making them particularly suitable to study aging and age-related diseases. On the other hand, the direct acquirement of a differentiated phenotype makes transdifferentiation not appropriate for more complex experimental model development, such as organoids (see below) which need proliferative cells to form. To overcome this limitation, a different procedure was developed to create by transdifferentiation neural progenitor cells (NPCs), an intermediate cellular population more appropriate to obtain a large number of proliferative cells.

## 5. Beyond 2D Models: Use of hiPSC-Derived Astrocytes for the Generation of 3D Models

The need of experimental models minimizing the amount of failures in the transfer of potential therapeutics from pre-clinical studies to clinic, pushed toward the development of 3D cultures. Obvious differences between 2D to 3D models mimicking live tissue are the limited contact with the surrounding cells and the extracellular matrix, the abnormal access to substrates and oxygen, the altered mechanical forces acting on cells, and only partial chemical contacts with adjacent cells. All these occurrences can alter the cellular maturation status [183]. The advancement from 2D to 3D cultures took advantage from the “self-organization” capability of hiPSC and ESC giving rise in non-adherent conditions to embryoid bodies, spheroids, and organoids (Figure 9). Spheroids are clusters of cells initially sticking each other without needing a scaffold; organoids should meet two criteria: comprising the cell types of the original organ, and being spatially organized similar to the original organ [184]. To improve the quality of organoids new technological advancements have been made to synthetize the biomaterials suitable for 3D scaffolds. In parallel, a different technology aimed to the same goal gave rise to the so-called organ-on-a-chip [185]. Briefly, an organ-on-a-chip is a structure shaped in chambers and micrometric channels, in which cells are seeded to form a homotypic or heterotypic 2D/3D culture. Often chips also have a microfluidic capability to emulate circulation and diffusion of biological components. In this section, we will provide a general view of the application of these technologies to reproduce and study astrocytes structural and functional interactions in specific brain sub-compartments that are relevant to LD pathogenesis, as the multicellular complexes supporting myelination, the BBB and the NVU (Figure 9). A pioneering procedure to obtain an organoid modeling of brain development starting from hiPSC was documented by Lancaster and colleagues [186]. The study was focused mainly on the neuronal component, but it indicated the possibility of reproducing in vitro the complexity of the whole human brain. In a first step they developed neuroectoderm from embryoid bodies. Then neuroectoderm was allowed to grow in a 3D culture and then embedded in a matrix of Matrigel as a scaffold allowing the further growth. The developing “brain” was transferred in a spinning bioreactor for up to 30 days. After this period, they were able to observe the disappearing of pluripotency markers (OCT4 and NANOG) and the increased expression of mature neural markers (SOX1 and PAX6), confirming a successful neural development. They also reported the development of brain regionalization, with forebrain and hindbrain marker expression, and the formation of discrete regions, which included the dorsal cortex, prefrontal cortex, hippocampus, and retinal structures as revealed by immunostainings. Interestingly, these structures also reproduced the orientation of radial glia in the ventricular zone, similar to the one described in the human brain (and that differs from the rodent one), as an additional validation of this model to recapitulate a miniaturized human brain. Finally, through calcium imaging, they could record tetrodotoxin-sensitive glutamate-induced calcium activity, further confirming the functional capability of these organoids. Other procedures were proposed at that time, but the one described above had the advantage to obtain later events of human cortex development, such as distinct cortical layers and outer radial glia. Relevant contributions to the development of 3D models starting from hiPSC and focusing on the neuroglial components are due to the group of Pasca [187,188]. They obtained spheroids reproducing a 3D cortical-like structure from human hiPSC in non-adherent conditions and without the use of scaffolds. In appropriate growing conditions they described the development of a laminated structure resembling the cortex and including mainly neurons, which they called human cortical spheroids (hCSs). They can assist to the later appearance of an increasing population of resting astrocytes (GFAP positive cells), capable of responding to the appropriate stimulus by acquiring reactive astrocyte markers (GFAP, vimentin and Lipocalin-2). In a following study, they focused on the astrocytic component obtained through the dissociation of hCS and immunopanning with the antibody for the astrocytic adhesion molecule GlialCAM. The astrocytic maturation analysis performed by RNA-seq revealed the disappearance of proliferative markers and the acquisition of an adult set of gene such as those expressing AQP4 and the high-affinity glutamate transporter EAAT1. Functional properties of single astrocytes isolated from spheroids and kept in culture indicated that these cells were able to uptake glutamate, promote synaptogenesis, and neuronal calcium signaling. As expected, the capability to phagocytose synaptosomes was lost in astrocytes from more mature spheroids. In summary, the 3D model developed by Pasca’ s team closely recapitulated the in vivo astrocytic development and their contribution to synaptic development. A different approach was proposed in Krencik’s paper [189], where they described the generation of 3D organoid-like structures fabricated by co-culturing pre-differentiated astrocytes and pre-differentiated neurons from hPSC. Although the method did not reproduce the full complexity of the native tissue, it offers the advantage to study the astrocyte-neuron relationships without biases due to other cell types. Moreover, it can be envisaged its utilization in exploring the physiopathological relationship between astrocytes and other glial cells. Since the earliest attempts to create cortical 3D models, oligodendrocytes, and their precursors (OPCs) were rarely observed, in keeping with the lack of differentiation factors and not yet completed neuronal maturation. More recently, a method to obtain spheroids enriched of oligodendrocytes was established [190], by using the appropriate ensemble of growth factors and bioactive compounds to promote OPC proliferation and survival. In these spheroids they can follow the maturation from precursors to mature oligodendrocytes capable of forming lamellae of myelin around axons, through stages recapitulating those observed in primary cells. The presence of astrocytes also made these spheroids potentially capable of reproducing the role of these cells in the control of myelination, and the model suitable to study the astrocyte contribution to myelin diseases. In line with the potential utilization in the study of oligodendrocyte pathologies, they analyzed the stage-dependent expression of genes associated with LDs such as Aicardi–Goutières syndrome, metachromatic leukodystropy and Krabbe disease. At present, other valuable methods to grow 3D models of myelination are available [191,192,193,194] and their heterocellular composition make them suitable for studies on the astrocyte-mediated LDs. In an exemplifying study, they describe the use of hiPSC from the LD Pelizaeus–Merzbacher disease patients to grow cortical spheroids [191].

BBB is another important multicellular structure where astrocytes exert an essential role that is affected in LDs (Figure 1). In particular, the vessel side (lumen) of BBB is composed of endothelial cells, the brain side of pericytes and astrocytic end-feet, and both sides contacting a basal membrane. The very low permeability of BBB to water and solutes relies on the presence on endothelial cells of a very efficient ensemble of tight junctions, which allow only the trans-cellular passage of molecules thanks to membrane transporters, and that distinguishes the BBB from the vessel wall of other tissues. The loss of BBB integrity and function causes several pathological conditions. In particular, the loss of function of the astrocytic component of BBB is thought to be one of the causes of the rare LD MLC. The classical transwells, which have been largely used in the study of BBB, lately gained benefit from the introduction of iPSC technology and from the development of sophisticated electrospun nanofibers but also from biocompatible and inexpensive paper as scaffold materials growing BBB cell types [195,196]. In parallel, 3D models of BBB take advantage of organ-on-a-chip technology with microfluidic capability, where hiPSC-derived cells can be grown and develop a BBB-like structure. From the first procedure for obtaining from human hiPSC, endothelial cells with features mimicking those of BBB [197], many studies followed [198,199,200,201]. Since then, the main common principle of the different procedures has been the need of co-culturing endothelial cells together with astrocytes and pericytes, to drive endothelial cells toward a BBB phenotype, featuring close fitting tight-junctions and the appropriate transporters. Organ-on-chip technology provides the advantage of allowing the appropriate aggregation of the cells mimicking BBB in brain vessels on the sides of a porous membrane with astrocyte end-feet laid on the endothelial layer. Taking advantage of the microfluidic, the shear forces created by the flow of solution reproduces brain vessels conditions. In recent years, BBB models on chips were used for studying the permeability of drugs and the contribution of BBB to brain diseases.

The NVU, consisting of neurons, glial cells, vascular cells (endothelial cells, pericytes, and vascular smooth muscle cells) along with the surrounding extracellular matrix, is a functional extension of the BBB where astrocytes exert the fundamental role to connect peripheral blood circulation with the brain neuronal activity. Disruption of the NVU underlies the development and pathology of multiple neurological disorders. Several methods to develop robust and translatable models of the NVU have been established (for an exhaustive review see Geoffrey Potjewyd et al. [202] and reference therein). These include transwell models, hydrogel models, 3D-bioprinting, microfluidic models, and organoids (Figure 9). By incorporating hiPSC-derived neural cells, these NVU models can provide a platform to study disease mechanisms, and the effect of astrocyte/BBB targeted drugs on neurons and neural circuit functionality.
Figure 9The figure offers a basic schematic representation of the progresses of the in vitro experimental models to study astrocyte role in LDs and other brain diseases. (**2D**) In the classical 2D models, cells dissociated from the original tissue are grown in monotypic (one cell type) cell cultures, or in co-cultures (two cell types mixed in culture plates or separated in transwell systems), which allow the study of astrocytes specific roles and their functional relationships with neurons, oligodendrocytes and endothelial cells. (**3D**) Taking advantage of the pluripotency and proliferative potential of hiPSC, the first 3D models (spheroids) were grown relying on staminal self-assembly capability in non-adherence conditions. The evolution of biotechnology, the advancement in media composition with the introduction of specific ensemble of growth factors and inducers/inhibitors, made it possible to grow organoids containing all the cell types needed to reproduce the original tissue structure. In parallel, the bioengineer evolution of components such as new scaffold materials, chips, microfluidic components allowed the creation of a multitude of microfluidic chips, which are largely used in the BBB/NVU studies. Further models to study astrocyte pathophysiology might include brain assembloids (combination of organoids representing different brain regions) where studying myelin features in the inter-regional connections, and brain organoids including hiPSC-derived endothelial cells useful for BBB/NVU studies. Organoids taking advantage of the microfluidic technology applied to chips are also implemented and used for NVU studies.
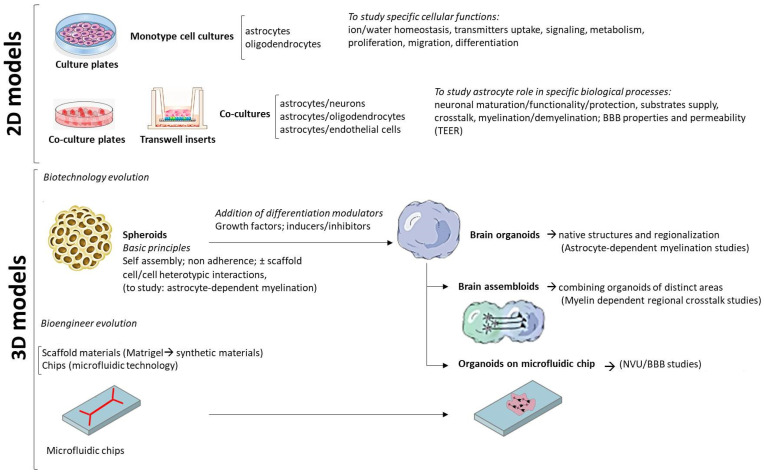


## 6. Advantages and Limits of hiPSC-Technology for LD Studies: From Developmental to Ageing Models

The introduction and progresses of the hiPSC technology has been of great help in particular for those fields suffering for the limitation of availability of healthy/diseased tissues, such as the rare neurological diseases. The main advantages offered by the use of hiPSC-derived cells, are the possibility of modeling these diseases (also in their developmental stages) [203,204,205], using the ensemble of cells potentially relevant for the pathology (otherwise not accessible) and reproducing the specific genetic platform of the single individual, including the multigenic mutation background (Figure 10). This technology is rapidly expanding and, hopefully, it will contribute to improve the efficacy of the in vitro research in the development/or repurposing of effective drugs for human use, not failing when exposed to clinical trials, because of low efficacy or unexpected toxicity. However, some limitations/difficulties must be considered in working with hiPSC. One of them is the possible the clonal heterogeneity of the stem cells derived from patients and healthy control (often also not age-matched) that can potentially mask some pathological phenotype when studying genetic diseases. The recent development of the CRISPR/Cas9 technique for DNA editing allowing the generation of isogenic lines, can overcome these issues. The poor capability of hiPSC-derived cells to reach a mature state, is another possible limitation that must be evaluated also during astrocyte/oligodendrocyte differentiation procedures. Moreover, hiPSC revealed incapable of maintaining the features acquired with aging by the original individual from whom starting cells, usually fibroblasts, were obtained [177,183,203]. The energetic metabolism, that is the cell production of the energy carrier ATP from sugars, amino acids and lipids, is tightly linked to the cellular developmental stage. Generally, proliferating cells (e.g., stem cells and cancer cells) rely on glycolysis, because it allows the use of substrates also for production of molecules needed for growing. On the contrary, differentiated cells rely mostly on mitochondrial Krebs cycle and respiratory chain [204]. Throughout reprogramming from fibroblasts to hiPSC, cells go through a shift of cell metabolism, from a balanced mitochondrial/glycolitic metabolism, proper of fibroblast, to a glycolytic metabolism, together with a decline of the mitochondrial compartment. When hiPSC are redirected to the final cells of interest, the energetic metabolism of hiPSC goes through a reverse switch from glycolytic to the metabolism proper of the final cell [204], which might be hampered by the previous mitochondrial decline experienced during reprogramming from fibroblasts to iPSC. This limitation applies especially to monotype cultures, co-culturing neural cells (astrocytes/neurons) is already an efficient strategy to overcome this limitation, and even more when culturing in 3D [183]. Aging goes along with a decrease in methylation of most of the genome, together with a hyper-methylation of a few discrete sights. The evolution of this epigenetic feature is lost with reprogramming to hiPSC, so rendering hiPSC-based cultured cells is not very suitable to represent reliable aging in models of aging diseases [177,203].

## 7. Conclusions and Future Perspectives

Over the last decades, the fundamental role of astrocytes in the onset and progression of neurological diseases, including rare LDs, has emerged. Considering the low availability of pathological tissue and the not complete reproducibility of LD phenotypes in the animal models, the hiPSC represent a powerful tool to achieve in vitro models where investigating LD pathological mechanisms. These studies will also add new knowledge on astrocyte-mediated mechanisms supporting myelin formation and maintenance. Indeed, LDs caused by astrocyte genetic defects have become a unique model for primary astrocytopathies, providing the opportunity to unravel astrocyte contribution to the pathogenesis of other demyelinating diseases. Although hiPSC technology will require further optimization to better define the subtypes and developmental stages of the differentiated astrocytes, in view of their potential as pharmacological targets, iPSC-derived astrocytes will be of extreme interest for the creation of drug screening and therapy testing platforms for a group of diseases for which no cure is currently available.

## Figures and Tables

**Figure 1 ijms-23-00274-f001:**
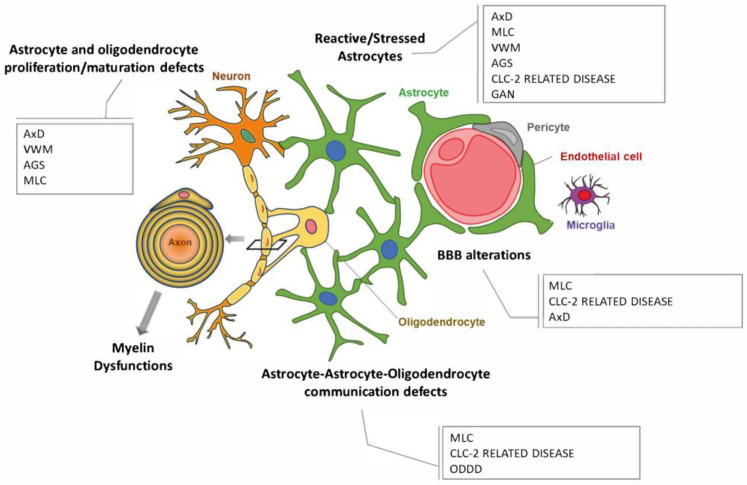
Schematic representation of astrocyte structural and functional defects found in the leukodystrophies caused by astrocyte dysfunctions.

**Figure 10 ijms-23-00274-f010:**
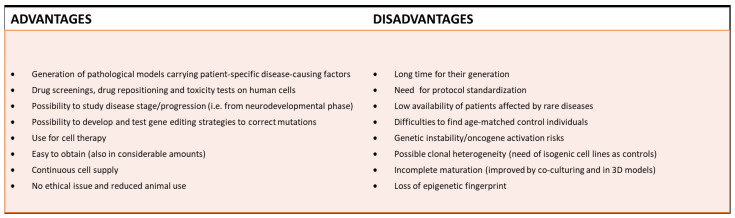
Advantages and disadvantages of hiPSC technology to study rare neurologic diseases.

**Table 1 ijms-23-00274-t001:** Differentiation protocols used to generate iPSC-derived astrocytes from patients affected by astrocytophaties.

*Patient-Derived Genetic Mutations*	*Astrocyte Differentiation/Maturation Procedures*	*% of Differentiated Cells*	*Expression of Astrocyte-Specific Markers*	*Ref.*
**Alexander Disease (AxD)**				
c.262C>T c.1246C>T	*Neural Induction Protocol* [55] *Astrocyte diff/maturation medium* DMEM/F12 + BMP4 + CNTF	80% GFAP^+^ at 6 months	GFAP, CNX43, AQP4, EAAT2	[56]
c.729C>T c.205G>A c.827G>T	*Neural Induction Protocol* [57] *Astrocyte diff/maturation medium* DMEM/F12 + EGF + FGF + heparin	90% S100β^+^ at 6 months	GFAP, S100β	[58]
c.729C>T c.249C>T c.218T>A	*Neural Induction Protocol* [59,60] *Astrocyte diff/maturation medium* Astrocyte medium + EGF + FGF + CNTF	80–90% GFAP^+^ at 120 days	GFAP, S100β, SOX9, CD44, EAAT1, GLUL	[60]
c.729C>T c.249C>T by CRISPR–Cas9 gene editing	*Neural Induction Protocol* [61] *Astrocyte diff/maturation medium* 1:1 DMEM/F-12 and Neurobasal + NAC + HB-EGF + CNTF + BMP4 + dbcAMP	80% GFAP^+^, S100β^+^ at 21 days	GFAP, S100β, ALDH1L1, EAAT1	[61]
**Vanishing White Matter (VWM)**				
c.1484A>G c. 806G>A	*Neural Induction Protocol* [62] *Astrocyte diff/maturation medium* N2B27-vitA + EGF + FGF2 + CNTF (for hWM astrocytes) or + 10% FBS (for hGM astrocytes)	80% GFAP^+^ at 35 days	GFAP, nestin, Id3, CD44, SOX9	[63]
c.1827_1838del c.1157G>A c.140G>A, c.1037T>C	*Neural Induction Protocol* [64] *Astrocyte diff/maturation medium* Astrocyte medium (Stemcell)	GFAP^+^ at 28 days	GFAP, S100β	[64]
**Megalencephalic Leukoencephalopathy with subcortical Cysts (MLC)**				
c.177+1g>t c.178-10t>a Del Ex10 (MLPA) c.423+1g>a/c.470C>A	*Neural Induction Protocol* [65] *Astrocyte diff/maturation medium* DMEM + FGF2 + CNTF	90% GFAP^+^ at 28 days	MLC1, GFAP, EAAT1, EAAT2, CNX43, vimentin, Kir4.1, S100β	Lanciotti et al.; manuscript in preparation
**Aicardi-Goutières Syndrome (AGS)**				
c.602T>A/p.V201D mutation in TREX1	*Neural Induction Protocol* [66] *Astrocyte diff/maturation medium* Astrocyte Medium Lonza	GFAP^+^ at 35 days	GFAP, S100β	[67]

Abbreviations: BMP4 (Bone Morphogenetic Protein 4); CNTF (Ciliary neurotrophic factor); EGF (Epidermal growth factor); FGF (Fibroblast Growth Factor); NAC (N-acetyl-cysteine); HB-EGF (Heparin-binding EGF-like growth factor); dbcAMP (Dibutyryl cyclic adenosine monophosphate); hWM (human white matter); hGM (human grey matter); GFAP (Glial fibrillary acidic protein); CNX43 (connexin 43); AQP4 (aquaporin 4); EAAT1/2 (Excitatory amino acid transporter 1/2); S100β (S100 calcium-binding protein beta); SOX9 (SRY-Box Transcription Factor 9); GLUL (glutamine synthetase); ALDH1L1 (Aldehyde Dehydrogenase 1 Family Member L1); Id3 (DNA-binding protein inhibitor of differentiation 3); Kir4.1 (inwardly rectifying potassium channel 4.1).

## Data Availability

Not applicable.

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
