# Peer review of "Human iPSC-Derived Astrocytes: A Powerful Tool to Study Primary Astrocyte Dysfunction in the Pathogenesis of Rare Leukodystrophies"

_ijms, 2021, doi:10.3390/ijms23010274_

Round 1

Reviewer 1 Report

Lanciotti and colleagues have comprehensively reported how to utilize human iPSC-derived astrocytes for the studies of rare leukodystrophies. The review started from the introduction of two populations of astrocytes, protoplasmic astrocytes and fibrous astrocytes. Then, they reported that clarifying the role of astrocyte in specific CNS diseases could help for therapeutic purposes. Authors tried to reveal the availability, advantages, future evolutions, perspectives and limitation of human iPSC-derived astrocytes to study pathologies implying astrocyte dysfunctions. I think it is relevant and interesting. 

The paper is well written, clear, and easy to ready, although some of the sentence structure can be improved. The figures are pretty and easy to understand, which I think it is not easy to find from other published material. 

They reported the pros and cons of the tools to study diseases related to astrocytes, using human iPSC. The conclusions are consistent with the arguments and evidence/references presented. 

Reviewer 2 Report

The review by Lanciotti et al., discussed the use of iPSC derived astrocytes in studying various leukodystrophies. The article discussed about different reported diseases and the animal and human models of the diseases. Adding a few more information and changing few details in the manuscript will improve the quality and will therefore be very beneficial to the readers.

  1. Using a mouse model of AxD (GFAPTg;Gfap+/R236H) reported decrease in classical protein constituents of myelin (Cnp, Mbp, Mog, Mobp, Mag, and Plp1). (Heaven et al.,2021)
  2. Li et al. study used GFAP mutations R79C,R239C and M73K mutation in the GFAP coding region This details should be discussed.
  3. Mutation in RNaseH2B and in TREX1 by Ferraro et al., should be discussed.
  4. Zhang et al., 2016 and Holmqvist et al., 2015 reported generation of GFAP::GFP astrocyte reporter lines from human adult fibroblast-derived iPS cells. Including this article will be beneficial for the readers.
  5. Huang et al., 2021 reported the development of 3D BBB model using hybrid paper/nanofiber-based cell culture platform studying the endothelial-astrocyte interaction. A discussion about this article will be beneficial for readers and may be included in the Figure 9.
  6. Correct the symbol of αβ-crystallin in Page 8 line 272 and spelling in Page 10 line 389.
  7. Define GFAP in Page 4 line 157.
  8. Include the mutations in iPSCs generated in each figure.
  9. In Figure 1 please try to add some indication that the myelin figure on left is a rotated slice on the axon on right. Also improving the clarity of the fonts in all figures will be needed. Being consistent with what colors used for each cell types through the entire manuscript figures will be better. Adding the references in the figures for each model will help the reader to access them easily.
  10. Please add reference to the various models in Figure 9.
